# Metabolomics and Proteomics in Prostate Cancer Research: Overview, Analytical Techniques, Data Analysis, and Recent Clinical Applications

**DOI:** 10.3390/ijms25105071

**Published:** 2024-05-07

**Authors:** Fatima M. Al-Daffaie, Sara F. Al-Mudhafar, Aya Alhomsi, Hamadeh Tarazi, Ahmed M. Almehdi, Waseem El-Huneidi, Eman Abu-Gharbieh, Yasser Bustanji, Mohammad A. Y. Alqudah, Ahmad Y. Abuhelwa, Adnane Guella, Karem H. Alzoubi, Mohammad H. Semreen

**Affiliations:** 1Department of Medicinal Chemistry, College of Pharmacy, University of Sharjah, Sharjah 27272, United Arab Emirates; u22103674@sharjah.ac.ae (F.M.A.-D.); u17100016@sharjah.ac.ae (S.F.A.-M.); u17103371@sharjah.ac.ae (A.A.); htarazi@sharjah.ac.ae (H.T.); ahmedm@sharjah.ac.ae (A.M.A.); 2Research Institute of Medical and Health Sciences, University of Sharjah, Sharjah 27272, United Arab Emirates; welhuneidi@sharjah.ac.ae (W.E.-H.); eabugharbieh@sharjah.ac.ae (E.A.-G.); ahmad.abuhelwa@sharjah.ac.ae (A.Y.A.); kelzubi@sharjah.ac.ae (K.H.A.); 3Department of Basic Medical Sciences, College of Medicine, University of Sharjah, Sharjah 27272, United Arab Emirates; ybustanji@sharjah.ac.ae; 4Department of Clinical Sciences, College of Medicine, University of Sharjah, Sharjah 27272, United Arab Emirates; 5School of Pharmacy, The University of Jordan, Amman 11942, Jordan; 6Department of Pharmacy Practice and Pharmacotherapeutics, College of Pharmacy, University of Sharjah, Sharjah 27272, United Arab Emirates; malqudah@sharjah.ac.ae; 7Department of Clinical Pharmacy, Faculty of Pharmacy, Jordan University of Science and Technology, Irbid 22110, Jordan; 8Nephrology Department, University Hospital Sharjah, Sharjah 27272, United Arab Emirates; adnane.guella@uhs.ae

**Keywords:** prostate cancer, omics, metabolomics, proteomics, clinical applications, biomarkers, analytical techniques

## Abstract

Prostate cancer (PCa) is a significant global contributor to mortality, predominantly affecting males aged 65 and above. The field of omics has recently gained traction due to its capacity to provide profound insights into the biochemical mechanisms underlying conditions like prostate cancer. This involves the identification and quantification of low-molecular-weight metabolites and proteins acting as crucial biochemical signals for early detection, therapy assessment, and target identification. A spectrum of analytical methods is employed to discern and measure these molecules, revealing their altered biological pathways within diseased contexts. Metabolomics and proteomics generate refined data subjected to detailed statistical analysis through sophisticated software, yielding substantive insights. This review aims to underscore the major contributions of multi-omics to PCa research, covering its core principles, its role in tumor biology characterization, biomarker discovery, prognostic studies, various analytical technologies such as mass spectrometry and Nuclear Magnetic Resonance, data processing, and recent clinical applications made possible by an integrative “omics” approach. This approach seeks to address the challenges associated with current PCa treatments. Hence, our research endeavors to demonstrate the valuable applications of these potent tools in investigations, offering significant potential for understanding the complex biochemical environment of prostate cancer and advancing tailored therapeutic approaches for further development.

## 1. Introduction

Cancer is a multifaceted and heterogeneous disease characterized by the uncontrolled growth of specific cells, which can originate from any cell type, disrupt normal proliferation, and lead to abnormal growth and multiplication [1]. According to recent statistics, prostate cancer (PCa) emerged as a prevalent form of cancer among men in 2023, constituting 29% of all new cancer diagnoses in men. Approximately 288,300 new cases of PCa were estimated, with around 34,700 resulting in fatalities [2]. PCa stands as a leading cause of cancer-related deaths in Western countries, primarily affecting men aged 45 to 60 [3,4]. The diagnosis of PCa typically involves methods such as prostate biopsies, Prostate Specific-Antigen (PSA) testing, Digital Rectal Examinations (DREs), Magnetic Resonance Imaging (MRI), or health screenings. Various risk factors, including family history, ethnicity, age, obesity, and environmental factors, collectively contribute to the pathogenesis of PCa [4,5,6]. Contrary to common belief, prostate cancer often progresses asymptomatically, with symptoms typically manifesting only in advanced stages, predominantly due to metastases. However, benign prostatic enlargement can cause symptoms such as frequent urination, nocturia, an urgent need to urinate, and urinary hesitancy, which are often erroneously associated with prostate cancer. These symptoms arise from an enlarged and inflamed gland compressing the urethra and the bladder, obstructing urine flow [5,7,8]. Therapeutic approaches for PCa vary depending on disease aggressiveness. Low-risk PCa patients can be recommended ‘watchful waiting’ or ‘active surveillance’ strategies, which involve monitoring without immediate treatment until progression necessitates palliative care. Intermediate and high-risk PCa often require curative intervention. The options include radical prostatectomy, radiotherapy, and hormone therapy. Radical prostatectomy and radiotherapy are the primary treatments for organ-confined PCa, often followed by androgen deprivation therapy (ADT), which is the first-line treatment for all men with high-risk or metastatic PCa to suppress androgen activity. Chemotherapy, particularly with taxanes like docetaxel and cabazitaxel, is necessary for recurrent or advanced PCa cases, improving survival outcomes for patients who have failed hormone therapy and making them standard first- and second-line chemotherapeutic agents [3,5]. However, current therapies often entail significant adverse effects, driving the quest for alternatives and highlighting the need for more precise, reliable, and objective methods. In response to these challenges, omics analytical approaches have emerged as promising tools for advancing PCa research and management. Genomics, transcriptomics, proteomics, and metabolomics are among the omics techniques utilized to detect genes, mRNA, proteins, and metabolites associated with PCa. These approaches not only aid in understanding disease mechanisms but also offer the potential to enhance PCa’s diagnosis, prognosis, and treatment outcomes [3,9]. Our review aims to highlight the significant impact of these tools on scientific research and their potential to address specific therapeutic challenges in prostate cancer. By focusing on the relevance of omics in PCa research and emphasizing its importance in improving diagnostic precision and treatment efficacy, we aim to provide valuable insights into the current state and future directions of PCa management.

## 2. “Omics” in Research

Histopathological tests are the current standard for cancer detection, but their subjective outcomes highlight the need for more accurate methods. Innovative therapeutic approaches are crucial for improving cancer treatment’s effectiveness. Omics analytical methods, including genomics, transcriptomics, proteomics, and metabolomics, have been developed to identify genes, mRNA, proteins, and metabolites, facilitating rapid progress in early disease detection. Each omics data type offers unique insights into underlying disease processes and the biological pathways impacted [9,10]. Combining omics technologies allows for a more comprehensive investigation of disease mechanisms [9,11]. Omics research is expected to address complex questions, aid in drug development, and improve diagnostics by comparing the changes and interactions of biomarkers within organisms. Challenges persist due to the complexity of these tasks [12]. Diseases have distinct biomarkers and metabolic profiles, with disease progression influenced by specific metabolic shifts. Identifying these biomarkers can reveal novel drug targets and lead to significant discoveries [13]. Diverse analytical techniques, such as Nuclear Magnetic Resonance (NMR), Liquid Chromatography–Mass Spectrometry (LC-MS), Gas Chromatography–Mass Spectrometry (GC-MS), and Matrix-Assisted Desorption Lazer/Ionization Time-of-Flight-Mass Spectrometry (MALDI-TOF-MS), are deployed to generate high-caliber data for identifying disease-specific metabolites, proteins, and underlying dysregulated pathways [14].

Metabolomics, a rapidly evolving field, falls within the comprehensive domain of the interdisciplinary ‘omics’ disciplines. It involves quantitatively analyzing intricate metabolic reactions within living biological systems and responding to genetic modifications or pathophysiological triggers [15]. Metabolomics enables the detection, identification, and measurement of small molecules with low molecular weights (up to 1500 Da) in a wide range of cells, tissues, and biological fluids, contributing to our understanding of various biological processes and disease mechanisms [16,17,18]. Proteomics, on the other hand, delves into the comprehensive exploration of proteins within biological contexts, going beyond mere identification to uncover their functions, structures, post-translational modifications (PTMs), interactions, and contributions to various cellular processes. An integral aspect of proteomics involves investigating post-translational modifications to uncover their functional significance. Moreover, proteomics aids in revealing the spatial and temporal organization of the proteome, providing insights into how living systems respond to different stimuli, diseases, or environmental factors. It also furnishes us with data about the precise location of proteins within cells or tissues, vital for comprehending their roles in cellular processes and identifying potential targets for therapeutic drugs. This wealth of information is invaluable for advancing our knowledge of cellular biology, disease mechanisms, and the development of targeted treatments [9,19]. These metabolites and proteins represent the end products of the biological hierarchy, as shown in Figure 1, originating from the genome (active genes), progressing through various gene transcripts (transcriptome), and culminating in proteins (proteome) and metabolites (metabolomes). This integrative “omics” approach can enhance PCa’s diagnosis, prognosis, and treatment by identifying the critical molecular mechanisms within various pathways.

## 3. Workflow of Metabolomics and Proteomics

While metabolomics and proteomics generally share a standard methodology, they differ in their specific approaches. Figure 2 provides an overview of the workflows of both fields. Metabolomics studies primarily utilize two approaches: targeted and untargeted metabolomics, based on whether the experimental setup permits the examination of a definite or indefinite group of metabolites. As part of recent research methods, targeted techniques require prior knowledge of the specific metabolites associated with a biological process [9,20].

### 3.1. Targeted Metabolomics

Metabolomics plays a vital role in understanding the origins of diseases, exploring metabolic pathways, and investigating the effects of factors like environmental changes, diseases, drugs, and genetics. Its significance is particularly notable in cancer research and clinical oncology, contributing to biomarker discovery and drug development [21,22]. In cancer research, metabolic profiling allows for comparisons between patients and healthy individuals, tracks disease progression, and identifies relevant biomarkers. Clinical oncology provides insights into treatment responses, predicts cancer risk, and assesses the likelihood of recurrence [21,22]. One approach in metabolomics is the targeted approach, like metabolite target analysis, which involves the use of specialized analytical techniques to detect and quantify a specific set of metabolites (target metabolites). This strategy is primarily employed for screening and investigations requiring high sensitivity [23,24]. It is also used to study a specific panel of metabolites or pathways strongly associated with a particular disease or condition of interest. In targeted metabolomics, the metabolites of interest are known in advance, and their quantification requires chemical standards. Sample preparation is tailored to retain only the relevant metabolites. Once these metabolites are extracted from a biological sample, an appropriate analytical system, such as mass spectrometry, chromatography, or NMR, is used for their detection, identification, and characterization. This process often involves referencing metabolomic databases like METLIN, the Human Metabolome Database (HMDB), and MetaboAnalyst. Following this analytical phase, statistical methods are employed, with principal component analysis (PCA) being a frequently utilized approach for identifying the biomarkers linked to particular biological functions or pathways [25]. Targeted metabolomics thus emerges as a powerful approach within the broader field of metabolomics for investigating pharmacokinetics in drug metabolism, as well as assessing the effects of therapeutics or genetic modifications on specific enzymes [9,16,21,24].

### 3.2. Untargeted Metabolomics

Unlike targeted approaches, untargeted methods offer the potential to discover novel and unique metabolites without prior information or predefined targets. This approach impartially measures a wide spectrum of metabolites from various biological samples [20]. One can achieve this through metabolic fingerprinting or metabolic profiling [21,26]. Metabolic fingerprinting reveals the overall metabolic pattern of a cell, tissue, or organism without prior knowledge of the specific metabolites involved, enabling the characterization of a biological system under particular conditions [21,27]. This approach facilitates sample classification and distinguishes specimens based on different biological states, such as the presence or absence of disease or pre- and post-treatment scenarios, by capturing unique metabolic patterns [21,28] Furthermore, metabolic fingerprinting is more applicable to running diagnostic tests and clinical routines [16,24]. In contrast, metabolic profiling quantifies the concentrations of all detectable metabolites in a biological sample, providing comprehensive biochemical information by identifying the metabolites and metabolic pathways associated with specific physiological or pathological conditions. Notably, the number of quantifiable molecules in metabolic profiling is usually lower than those contributing to metabolic fingerprinting, and primarily so in urine samples (e.g., less than 50% are found in urine). Therefore, metabolic fingerprinting is better suited to sample classification and building statistical models [21]. Following metabolite identification, the interpretation phase seeks to establish connections between these significant metabolites and crucial pathophysiological processes or pathways. Any potential biomarkers discovered in metabolomics studies undergo additional validation through clinical trials or field studies [9]. 

### 3.3. Proteomics Workflow

The proteomics workflow exhibits specific differences in its methodology compared to metabolomics, although both often share a common approach. Proteomics primarily relies on liquid chromatography coupled with mass spectrometry (LC-MS), a widely used method for analyzing live single cells with minimal sample requirements and high sensitivity. LC-MS is categorized into two main approaches: bottom–up and top–down methods. The widely adopted bottom–up technique, known as “shotgun proteomics”, is favored for its extensive proteome coverage. It involves the enzymatic digestion of proteins into peptides after sample extraction. Subsequently, multidimensional LC separation and tandem MS (MS/MS) characterization are conducted, followed by protein identification using statistical databases [5,9]. In contrast, the top–down approach first separates protein mixtures and then sequences intact proteins, allowing for the identification and quantification of individual post-translational modifications. Top–down proteomics prefers the use of electron spray ionization (ESI) due to its ability to generate multiple charged precursor ions, which facilitates the effective dissociation of large protein ions and provides more MS/MS opportunities compared to matrix-assisted laser desorption/ionization (MALDI), which primarily generates singly charged species [9,29]. For targeted proteomics, selected reaction monitoring (SRM) and Multiple Reaction Monitoring (MRM) address sensitivity and reproducibility limitations, enabling the quantification of numerous proteins. Absolute quantitation is achieved by spiking peptide mixtures with labeled standards. These targeted approaches are often used alongside shotgun methods. The data generated from these analyses can be accessed through proteome repositories, facilitating bioinformatics data mining and analysis [5,9,11,29].

#### Validation of Quantitative Proteomics Data

The validation of chosen biomarker candidates from the discovery phase typically involves targeted MS-based assays or targeted proteomics analyses. Targeted proteomics employs the measurement of candidate biomarker peptides alongside their heavy-isotope-labeled synthetic counterparts, enhancing quantification accuracy and ensuring the precise measurement of the intended peptide with high specificity. Commonly utilized targeted MS techniques include selected reaction monitoring (SRM) on a triple quadrupole mass spectrometer and Parallel Reaction Monitoring on a high-resolution mass spectrometer (e.g., Q-Exactive). These targeted MS assays offer high accuracy, selectivity, and sensitivity by employing a two-stage mass filtering of both precursor and fragment ions with high resolution. Recent advancements in MS technology enable the large-scale validation of candidate biomarkers involving hundreds of peptides [30]. Furthermore, as a validation method, Western blotting offers a snapshot of dataset quality, relying on antibody reactivity to a specific antigen. It is more beneficial for confirming specific biological findings from the entire dataset than for abundant proteins alone. Other validation techniques may also be considered, such as immunohistochemical staining during microscopy or targeted mass spectrometry. While these approaches may be time-consuming and costly, they provide valuable information, including detailed protein localization and the relative abundance of proteins within specific cellular structures [31].

## 4. Exploring Biomarkers for Prostate Cancer Detection

Cancer biomarkers are typically categorized into prognostic, predictive, and pharmacodynamic markers. Prognostic markers offer insights into the expected course of cancer and its likely outcome, guiding treatment decisions. Predictive markers evaluate the potential benefits of specific treatments, while pharmacodynamics markers assess a drug’s immediate impact on a tumor, aiding in dosage determination during early anticancer drug development [32]. To explore novel biomarkers for prostate cancer detection, researchers have undertaken multi-omics studies focusing on prostate cancer due to its distinct characteristics and dysregulation of various associated pathways. These studies utilize patients’ urine, seminal fluid, and blood plasma/serum samples. While urine and seminal fluid contain a richer array of metabolites, urine collection is more convenient due to its simpler process [13,21,33,34]. A unique set of biomarkers has been identified based on their identifiability at the protein level and their detectability in biological fluids. These biomarkers include metabolites like dihydroxybutanoic acid, xylonic acid, and pyrimidine, which are detected using mass spectrometry and liquid chromatography techniques [21]. These explorations involved comparing benign and malignant prostate tissues, revealing promising prospects. While certain limitations impact the levels and our ability to differentiate between aggressive and non-aggressive cancer types, the spotlight remains on the widely utilized prostate-specific antigen (PSA), which plays a significant role in managing prostate cancer; it is detected and quantified using Liquid Chromatography-Tandem Mass Spectrometry (LM-Tandem MS) [35]. Other biomarkers like human glandular kallikrein 2 (hK2), annexin 3 (ANAX 3), beta-2-microglobulin (β2M), Microseminoprotein-beta (MSMB), serum amyloid A (SAA), and Engrailed-2 (EN2) are also being explored and detected by different proteomic techniques, most importantly by LC/MS, which is a robust method for characterizing proteins [36]. PSA, an androgen-dependent serine protease, has various applications in PCa management. However, elevated levels might not always indicate cancer, and the threshold of 4.0 ng/mL is debated due to its inadequate sensitivity and specificity [36,37]. hK2, a serine protease like PSA, aids in converting pro-PSA to active PSA. It is considered alongside PSA, but cross-reactivity issues exist [36,38]. Annexins, calcium, and phospholipid-binding proteins have complex roles in cell processes and tumorigenesis, but their consistency as PCa biomarkers is debated. ANAX 3 has potential as a tissue and urinary biomarker [36,39]. β2M is a low-molecular-weight protein associated with various cancers, including PCa, and may help distinguish BPH from PCa [36]. MSMB, produced with PSA from the prostate gland, decreases with PCa’s progression, possibly acting as a tumor suppressor. Earlier studies show lower serum and urinary MSMB in PCa compared to healthy controls [40]. SAA, a liver-produced lipoprotein, has implications for inflammation and cancer and is being explored for its prognostic and therapeutic value in various cancers, including PCa [41]. EN2, from the HOX gene family, has been studied as a urinary biomarker potentially indicating PCa volumes [36]. Furthermore, other biomarkers are considered important in detecting the presence of PCa. Metabolic biomarkers such as myoinositol, citrate, polyamine spermine, sarcosine, kynurenine, choline, proline, leucine, and uracil found in the prostatic fluid indicate prostate cancer. These markers are best identified using MS techniques (GC-MS, LC-MS) or NMR; they aid in the diagnosis and prediction of disease progression, shedding light on the molecular mechanisms behind prostate tumor growth [42]. Sarcosine, a derivative of the amino acid glycine, emerges as a promising non-invasive biomarker in prostate cancer due to its substantial increase in urine, tissue, and plasma during advanced stages of PCa, including metastasis. A previous study investigating the sarcosine levels in serum revealed its superior predictive efficacy compared to total PSA and free PSA in detecting PCa among patients with a total serum PSA < 4 ng/mL. Additionally, serum sarcosine exhibited the largest area under the curve (AUC) for predicting low-grade, low-PSA PCa, suggesting its potential in diagnosing PCa among individuals with normal PSA levels and identifying candidates suitable for non-aggressive therapies and active surveillance [43]. Various methods are used to analyze the sarcosine in biosamples, primarily involving derivatization combined with mass spectrometry, including GC-MS and LC with tandem MS for urinary sarcosine [44]. Intracellular sarcosine levels exhibit marked elevation in invasive prostate cancer cell lines compared to benign prostate epithelial cells, indicating its role in cancer progression [45,46]. Skreekumar et al. utilized the Oncomine Concept Map tool to identify increased methyltransferase activity in metastatic samples, implying a connection between altered sarcosine levels and modifications in the biochemical pathways linked to the progression of prostate cancer to more advanced stages [47]. Stabler et al. discovered that serum methionine metabolites are risk factors associated with the progression of metastatic prostate cancer. These findings strongly indicate a close relationship between changes in metabolic activity and prostate cancer progression [48]. Montrose et al. analyzed the metabolic changes in colorectal tumors induced by azoxymethane, revealing elevated sarcosine levels and an increased expression of GNMT and DMGDH, the enzymes responsible for producing this metabolite [49]. Furthermore, Dahl et al. documented, for the first time, the involvement of sarcosine in regulating the oncoprotein HER2/neu in androgen-dependent prostate cancer cells [50]. Another study revealed a notable elevation in serum sarcosine levels among individuals with metastatic disease. Additionally, this biomarker was identified as a risk factor for progression and survival in chemotherapy-treated patients with metastatic castration-resistant prostate cancer [51]. Moreover, Khan et al., employing both in vitro and in vivo preclinical models, confirmed sarcosine’s role as an oncometabolite in prostate cancer [52]. Kynurenine, detectable in plasma, urine, and tissue, is another metabolite linked to prostate cancer’s development and has more recently been associated with cancer’s aggressiveness based on Gleason scores [45,53]. Liquid chromatography is the primary method for assessing kynurenine levels in the body [54]. Choline, implicated in cancer development and DNA repair modulation, can function as a predictive biomarker for prostate cancer when its levels exceed 10 mol/L, in contrast to healthy individuals [55]. The standard method for choline analysis involves the utilization of high-performance or ultra-high-performance liquid chromatography combined with MS detection, often referred to as HPLC/UHPLC-MS [56]. 

Additionally, exosomes, or extracellular vesicles (EVs), are emerging as crucial biomarkers for prostate cancer diagnosis, treatment personalization, and prognosis assessment [57,58]. The unique prostate-cancer-specific contents found in blood and urine EVs serve as biomarkers for prostate cancer and metastasis. Various proteins on exosomal surfaces, including prostate-specific antigen (PSA) and exosomal RNAs, particularly microRNAs (miRNAs), have shown diagnostic potential. Studies confirm the high diagnostic value of plasma-exosomal miRNAs and identify specific miRNAs as potential biomarkers of castration-resistant prostate cancer (CRPC) [57]. Notably, the exosomal androgen receptor splice variant (AR-V7) correlates with treatment response and poor prognosis in CRPC patients [59]. Extracted EV-RNA has advantages in its biomarker potential over traditional methods, with specific EV populations representing heterogeneous tumors. Exosomal communication also influences the tumor microenvironment, with exosomal miR-375 promoting osteoblast activity and exosomal αvβ3 integrin implicated in aggressive cancer phenotypes [60]. Moreover, recent studies, such as the one by Joshi S. et al., have underlined the significance of metabolite signatures from exosomes in predicting responses in breast cancer patients, suggesting that a similar approach could be employed for prostate cancer [61]. Furthermore, various proteins like survivin and HSP72 have been identified as potential biomarkers through exosome isolation methods. Urinary exosomes also hold promise as non-invasive biomarkers for prostate cancer’s diagnosis, prognosis, and monitoring [57]. The findings of the study by Gan et al. suggested that mRNA from urinary exosomes holds potential as a novel and non-invasive indicator for prostate cancer’s diagnosis and prediction [62]. Overall, exosomes represent a multifaceted avenue for improving prostate cancer management through their diverse biomarker potential across different stages of the disease [57,63,64,65]. For a comprehensive overview of the various biomarkers mentioned above, refer to Table 1, which provides valuable insights into their characteristics and potential applications.

## 5. Multi-Omics Analytical Techniques

Analyzing the diverse array of small-scale metabolites and proteins poses a significant challenge. However, multiple analytical platforms are available to tackle this challenge, each with its advantages and disadvantages, as outlined in Table 2. These platforms vary in terms of sensitivity, specificity, reproducibility, sample preparation requirements, and equipment costs. Therefore, the choice of technique depends on various factors, including the characteristics of the analyte, the type of sample, the objectives of the analysis, and the resources available in the laboratory [27,66]. Examples of these analytical methods include LC-MS, GC-MS, MALDI-TOF-MS, and NMR.

### 5.1. Liquid Chromatography-Tandem Mass Spectroscopy (LC-MSMS)

Liquid Chromatography-Tandem Mass Spectrometry (LC-MS/MS) is a technique used for analyzing prostatic tissues and biofluids which utilizes mass spectrometry (MS). MS techniques, when coupled with separation methods such as liquid chromatography (LC), have been leading advancements in biomarker innovation studies. The LC-MS/MS method offers superior sensitivity and precision for quantifying metabolites and proteins [18,67,68]. In this technique, the metabolites and proteins in a sample are initially separated through liquid chromatography, and then a mass spectrum is generated to identify these metabolites, proteins, and their intermediates [14,69]. The integration of MS with LC has significantly improved pharmaceutical development by enabling reliable quantitative and qualitative analyses of a mixture of compounds [70,71]. MS detects analytes based on their ionization and fragmentation, producing unique fragments sorted by their mass-to-charge ratios (*m*/*z*) [24]. It provides the molecular weights of various substances and some structural information based on their mass-to-charge ratio through different ionization methods, such as electron impact and chemical ionization, yielding a mass spectrum [71]. Advancements in ionization methods allowing for the integration of liquid chromatography with mass spectrometers have significantly boosted the use of LC/MS in metabolomics and proteomics. This enables the identification of many polar and non-polar compounds, estimating the concentrations of unidentified analytes without the need for a derivatization step and using minimal sample amounts. High-performance liquid chromatography (HPLC) is a separation technique employed in LC/MS. It involves injecting a sample mixture (analyte), under high pressure, into a solvent flow (mobile phase) through a column filled with microscopic particles (stationary phase) [72]. The separation process relies on the nature of the stationary phase, causing analytes in the sample mixture to elute according to their affinity with either phase. This produces peaks in a chromatogram, indicating different analytes, their retention times, and their area under the curves (AUC). Normal-phase liquid chromatography (NP-LC) retains hydrophilic analytes more than hydrophobic ones, as its stationary phase is more polar than its mobile phase. In contrast, reversed-phase liquid chromatography (RP-LC) exhibits the opposite behavior [73]. RP-LC coupled with mass spectrometry (RP-LC/MS) using C18 columns is widely used for global metabolomics and proteomics as it separates hydrophilic and hydrophobic molecules. However, RP columns have limitations in their retaining of ionic or highly hydrophilic molecules. On the other hand, incorporating silicon hydride (Si-H) groups as an alternative to silanol (Si-OH) groups and using cation/anion ligands in RP C18 columns has greatly enhanced the retention of polar molecules, improving metabolome coverage. Hydrophilic-interaction liquid chromatography (HILIC) is also gaining prominence as an alternative to HPLC in metabolomic studies. HILIC utilizes a water-based mobile phase and normal-phase columns to effectively separate ionic and highly polar compounds, preventing their elution in the void volume [24,73]. This approach overcomes the solubility challenges faced in normal-phase liquid chromatography (NP-LC) [74]. By combining different analytical methods to assess the metabolome comprehensively, proteomics is gaining popularity despite its cost and time implications [24]. Integrating LC/MS technologies into analyzing diverse biological samples has shown promise in identifying human disorders and studying various diseases like cancer [75]. LC-MS is increasingly being used to explore pharmaceutical compounds across multiple stages, from drug discovery to identifying impurities and degradation products [76]. It has proven effective in quantifying tissue-specific complexity and highlighting metabolic changes in response to anticancer drugs [77]. Alongside statistical analyses, LC-MS is a powerful and efficient diagnostic tool for human diseases [78]. 

### 5.2. Gas Chromatography–Mass Spectroscopy (GC-MS)

Gas Chromatography–Mass Spectrometry (GC-MS) is an analytical method frequently employed in research. This technique enables the analysis and isolation of substances that do not dissolve upon vaporization. It is particularly well-suited for compounds with relatively low molecular weights that are nonpolar or exhibit both characteristics, making it a favored choice for comprehensive drug screening. GC-MS typically employs electron impact ionization combined with full-scan mass detection, allowing for the separation, purity assessment, and identification of specific components within mixtures [24]. Unlike high-performance liquid chromatography (HPLC), which employs a liquid mobile phase, GC utilizes an inert carrier gas like nitrogen, hydrogen, or helium. The choice of carrier gas depends on the chemistry of the analytes and the preferred detection method. In GC, the stationary phase can be either solid (Gas/Solid Chromatography) or liquid (Gas/Liquid Chromatography) and housed in a metal or glass column. The sample (solute) is first dissolved in a solvent, vaporized, and then carried by a mobile phase into the column. The analytes’ solubility determines their movement speed within the column in the stationary phase. This results in a retention time as the mobile phase passes through the column, effectively separating components based on their partition coefficient and boiling point. The resulting chromatogram displays the analysis findings, with varying retention times for the substances in the sample. Over time, both GC and MS techniques have seen significant advancements, and their combination has proven to be a valuable approach to separation, detection, and identification in metabolomics and proteomics studies. In the context of metabolomics and proteomics, the targeted metabolites can be either thermally stable and volatile, such as ketones, aldehydes, and organic acids, or non-volatile, including amino acids, lipids, and amines. Analyzing non-volatile metabolites typically requires derivatization, which involves chemical processes like acylation, sialylation, alkylation, and esterification [12]. These processes enhance the volatility and thermal stability of non-volatile samples, making them easier to evaluate and reducing the polarity of non-volatile compounds to facilitate analysis. The sample can be injected in either a split or split-less mode during GC-MS analysis, depending on its concentration. Subsequently, the sample is vaporized and ionized using various methods, selected based on the physicochemical properties of the target analyte and the specific application used. Standard ionization techniques for GC-MS include electron impact, a frequent choice, and chemical ionization, among other options. The mass separator then isolates the samples, facilitating their detection through a mass spectrum [24].

### 5.3. Matrix-Assisted Laser Desorption/Ionization Time-of-Flight Mass Spectrometry (MALDI-TOF-MS)

Mass spectrometry is a powerful analytical method that involves the conversion of samples into charged molecules, followed by the measurement of their mass-to-charge ratio (*m*/*z*). In MALDI-TOF mass spectrometry, the ion source is matrix-assisted laser desorption/ionization (MALDI), and the mass analyzer is a time-of-flight (TOF) analyzer. This approach is characterized by its gentle ionization method, using a laser which vaporizes the analyte molecules from a matrix without causing fragmentation. It is particularly suitable for analyzing biomolecules like peptides, lipids, saccharides, and various organic macromolecules. MALDI-TOF mass spectrometry finds applications in numerous omics fields, including proteomics, metabolomics, lipid-omics, and glycomics, and it is precious for determining protein molecular weight, identifying protein sequences, characterizing protein structures, and quantifying protein contents [79]. When preparing a sample for MALDI-MS analysis, the sample is mixed or coated with an energy-absorbing organic compound known as a matrix. As the matrix dries and crystallizes, the sample becomes entrapped within it. A laser beam is then employed to ionize the sample within the matrix. This laser-induced desorption and ionization process generates singly protonated ions from the analytes in the sample. These protonated ions are accelerated at a fixed potential, causing them to separate based on their mass-to-charge ratio (*m*/*z*). Various types of mass analyzers, such as quadrupole mass analyzers, ion trap analyzers, and time-of-flight (TOF) analyzers, can be used to detect and measure charged analytes. During MALDI-TOF analysis, the *m*/*z* ratio of an ion is determined by measuring the time it takes to traverse the length of the flight tube. This information is then utilized to generate a characteristic spectrum known as a peptide mass fingerprint for the analytes in the sample. This resulting spectrum exhibits unique peaks with specific mass-to-charge values on the *x*-axis and intensity on the *y*-axis, and it can be compared to a database of spectra from known organisms [80]. MALDI-TOF-MS boasts several advantages, including rapid results (typically within 10 min), leading to faster and more accurate patient treatment. Furthermore, the “soft” ionization method used in MALDI-TOF allows for the observation of ionized molecules with minimal fragmentation, as the formed ions possess low internal energy [79]. The introduction of MALDI-TOF-MS has facilitated the discovery of disease-related biomarkers and significantly contributed to advancements in disease diagnosis and personalized treatment [79]. 

### 5.4. Nuclear Magnetic Resonance (NMR) 

NMR is widely utilized in metabolomics owing to its notable advantages, which include its high reproducibility, minimal sample preparation, and nondestructive nature [81]. It can analyze various samples, ranging from complex liquid mixtures to intact tissues, cell extracts, and solid samples of living organisms, through a high-resolution magic angle spinning (HR-MAS) technique. NMR operates based on spinning charged protons in a nucleus, generating a magnetic field from them. When an external magnetic field is applied, the spin of the nuclei aligns with or against the magnetic field, resulting in a resonance frequency proportional to the magnetic field strength. This process produces a characteristic signal for each resonating nucleus, such as 1H, 13C, 17O, P31, and 15N, which have odd atomic numbers and masses [12,24]. The phenomenon of chemical shift, influenced by the electron shielding around the nuclei, allows for the structural interpretation, quantification, and identification of metabolites and proteins using established references [24]. NMR technology has seen advancements, including higher magnetic fields (>900 MHz) and cryoprobe technology. 13C NMR spectroscopy offers a broader spectral range than 1H NMR despite its naturally low abundance, which limits its sensitivity. Cryoprobe technology involves cooling NMR detectors to cryogenic temperatures, reducing the signal-to-noise ratio and the time required for signal recording in 13C NMR spectroscopy [24]. In prostate cancer, NMR aids in the identification of valuable biomarkers and provides insights into their in vivo magnetic resonance spectroscopy (MRS) metabolic profile. It facilitates the investigation of the biochemical and metabolic changes associated with prostate cancer [82]. However, compared to other MS techniques, NMR is less sensitive and requires larger sample volumes, more extensive cell numbers, and expensive instrumentation. The adoption of alternative MS techniques is driven by the lower sensitivity of NMR spectroscopy [83]. In conclusion, NMR spectroscopy remains essential in metabolomics, despite its poorer sensitivity and resolution than mass spectrometry. It enhances our understanding of systems biology, aids in identifying biomarkers and therapeutic targets, and connects laboratory discoveries with practical applications. While complex mixture detection and quantification challenges persist, continuous initiatives to improve its sensitivity, resolution, and data gathering indicate significant progress. Advances in NMR offer the potential for enhancing biological understanding and aiding disease therapy [84].

**Table 2 ijms-25-05071-t002:** The advantages and disadvantages of the commonly employed analytical techniques.

Analytical Platform	Advantages	Disadvantages	References
LC/MS	Highly sensitive technique. Analyzes polar compounds of different weights based on an ionization method. No derivatization needed. Suitable for heat-sensitive compounds. Compatible with liquids and solids. Requires minimal sample volumes.	Destructive technique. Costly equipment requiring expertise. Subject to unwanted solvent matrix effects. Prolonged analysis duration (15–40 min/sample). Generates variable adducts based on compound nature. Not suitable for gases.	[9,12,85]
GC/MS	Quantitative, reproducible, and sensitive technique. Direct analysis of volatile compounds. Effective for analyzing mixtures and small hydrophobic organic and certain inorganic compounds. Compatible with gases and liquids. Generally, it is more cost-effective than LC-MS due to its simpler detector.	Destructive technique. Inappropriate for non-volatile and heat-sensitive compounds. Necessitates separation and derivatization, which can mask the results. Long analysis time (20–40 min per sample).	[9,12,85,86]
MALDI-TOF-MS	Rapid analysis. High sensitivity. Minimal sample preparation. High mass range.	Sample homogeneity. Matrix-related peaks. Costly equipment. Limited in Gas-Phase Ionization.	[79]
NMR	Quantitative non-destructive technique. No requirement for harsh sample treatment before or during analysis. A sole internal reference is sufficient for precise quantification of all spectrum metabolites. Facilitates bio-fluid and tissue analysis without separation or preparation. Rapid analysis (2–3 min/sample). Analyzes both liquids and solids. Derivatization is not required. Automation integrated.	Limited sensitivity. Costly equipment. Signal overlap from the absence of prior separation. Does not identify inorganic ions or salts. Cannot detect non-protonated samples. Needs large sample volumes (0.1–0.5 mL).	[9,12]

Abbreviations: GC-MS: gas chromatography–mass spectrometry, LC-MS: liquid chromatography–mass spectrometry, MALDI-TOF-MS: matrix-assisted laser desorption/ionization time-of-flight mass spectrometry, and NMR: nuclear magnetic resonance.

### 5.5. Validating Analytical Techniques Used in Omics: A Comparative Exploration of MS and NMR Methods

These various analytical techniques have distinct designs and prerequisites. For instance, LC/MS is optimal for analyzing thermolabile, non-volatile, and polar substances, while GC/MS is recommended for volatile and less polar compounds. Additionally, the specialized technique MALDI-TOF-MS provides a rapid and accurate analysis of various biomolecules, making it valuable in the ‘omics’ era. Both NMR and MS analyses share similarities in their being labor-intensive and potentially tissue-damaging. However, MS analysis offers higher sensitivity in uncovering metabolites and proteins, whereas NMR is an impartial and non-destructive method for real-time detection. Nevertheless, NMR’s sensitivity is limited due to the challenges in identifying low-abundance analytes and signal overlap, which hinders precise quantification and metabolite identification (See Table 3 for a comparison between MS and NMR) [12,24,25].

Analytical validation primarily focuses on assessing analytical processes, including intra- and inter-assay variability evaluations. These processes can involve single instrumental techniques such as MALDI-TOF or NMR or combinations of multiple instruments like LC-MS and GC-MS. The specific validation methods may vary based on the chosen techniques. Typically, the initial biomarker discovery phase utilizes high-performance equipment like LC-MS, LC-MS/MS, MALDI-TOF, or MALDI-TOF/TOF to create a molecular signature based on molecular weights. Subsequently, these molecular features are identified through database matching or more advanced fragmentation methods. Various approaches can be employed for conducting validation procedures [32].

## 6. Data Processing and Statistical Analysis in Metabolomics and Proteomics Studies

Metabolomics and proteomics are powerful omics technologies capable of generating vast amounts of data, often requiring intricate data processing and statistical analysis to derive meaningful biological interpretations. Common to both, sample preparation, data analysis, and database matching are essential for identifying and quantifying metabolites and proteins [29]. Researchers leverage various statistical methodologies and bioinformatics tools to unearth valuable patterns, biomarkers, or noteworthy discoveries from the extensive datasets produced during the analytical stages. In the following section, we will explore the essential aspects of data processing and statistical analysis in metabolomics and proteomics studies. 

### 6.1. Data Preprocessing

In metabolomics, data preprocessing is of the utmost importance due to the diverse analytical techniques used to collect raw data, each producing unique data formats. The critical steps in data preprocessing include normalization, alignment, and noise filtering. Normalization is employed to adjust for sample variations, ensuring comparability by eliminating concentration differences. Alignment is crucial for correcting the peak shifts caused by pH, temperature, or instrument variations, ensuring data accuracy. Noise filtering techniques, including baseline subtraction and smoothing, enhance signal-to-noise ratios while retaining essential information [25]. On the other hand, in proteomics, data preprocessing is equally critical. Mass spectrometry is a common technique for protein identification and quantification. In MS data analyses, proteins from a sample are extracted and digested using proteases to generate peptides. Additional enrichment and fractionation steps are introduced when dealing with complex samples or specific subsets of proteins/peptides. The obtained peptides are analyzed using liquid chromatography coupled with mass spectrometry (LC-MS) [29]. Proteomics preprocessing steps may include baseline correction, deconvolution, and peak picking. These steps enhance data quality by removing noise and artefacts, allowing for accurate protein identification and quantification [87].

### 6.2. Statistical Analysis

Statistical analyses in metabolomics employ both univariate and multivariate approaches. A univariate analysis involves the examination of individual metabolites, comparing their concentrations across different sample groups. When dealing with data involving two groups, whether in unpaired or paired analyses, we can conduct fold change analyses and *t*-tests and visualize the results using volcano plots (represented in Figure 3D). In the case of data with multiple groups, we can employ a one-way analysis of variance (ANOVA), subsequent post hoc analysis, and correlation analysis. As multi-omics data typically exhibit variations corresponding to phenotypes or experimental conditions, it is advisable to employ multivariate analyses that enable the simultaneous observation and analysis of more than two statistical variables. Multivariate analysis simplifies complex datasets, revealing the patterns and relationships among metabolites. The widely used techniques for dimensionality reduction and sample classification in multivariate analyses include principal component analysis (PCA) and partial least squares discriminant analysis (PLS-DA). These methods aid in identifying biologically relevant factors [25]. PCA (shown in Figure 3A) is frequently employed as a preliminary analysis and quality control step in metabolomics data to identify intergroup classification trends and detect any outlier points within the data. Metabolomics analyses demand powerful software with functions including the processing of raw spectral data, statistical analysis for significant metabolite identification, connecting to metabolite databases, bioinformatics analysis with network visualization, and the integration and analysis of multi-omics data. For example, MetaboAnalyst 5.0 is a comprehensive, freely accessible web-based metabolomics analysis platform that provides extensive online tools for metabolomics data analysis, statistical analysis, functional annotation, and data visualization. Additionally, it has the capability to generate heatmaps and conduct metabolic pathway analyses (illustrated in Figure 3) [25].

However, statistical analyses in proteomics share similarities with metabolomics. Univariate analysis assesses individual protein expression changes between sample groups using tests like an ANOVA or *t*-tests. In multivariate analysis, complex protein datasets are explored, often employing PCA, PLS-DA, or hierarchical clustering (Figure 4). These methods help identify the protein patterns associated with specific conditions or biological processes. In metabolomics and proteomics, challenges include dealing with high-dimensional data, selecting appropriate statistical tests, and addressing issues related to multiple testing corrections. Additionally, batch effects, missing data, and data normalization are critical to ensure accurate statistical analysis [25,87]. Data processing and statistical analysis are indispensable components of metabolomics and proteomics studies. They enable researchers to uncover biomarkers, identify pathways, and gain insights into biological mechanisms. As these omics fields continue to advance, robust and innovative data processing and statistical methods are vital in translating complex data into biologically meaningful knowledge [25,87].

## 7. OMICS Technologies in Prostate Cancer: Advancing Diagnosis, Treatment, and Personalized Medicine

OMICS technologies, encompassing metabolomics and proteomics, hold immense promise for transforming cancer research and addressing the global healthcare challenges of cancer. These advanced methodologies offer comprehensive insight into the underlying pathophysiology of cancer, unraveling the metabolic differences between healthy individuals and cancer patients. This knowledge is pivotal for enhancing cancer’s diagnosis, prognosis, and monitoring and establishing disease-specific profiles correlating metabolites with cancer aggressiveness. Moreover, OMICS technologies contribute significantly to identifying potential pharmacological targets, reducing our reliance on toxic medications [8,88]. Personalized treatment approaches, aligned with precision medicine principles, rely on identifying molecular abnormalities through metabolic profiling and phenotyping. Personalized medicine involves the exploration of novel biomarkers derived from various sources, necessitating validation in independent patient groups to confirm their clinical significance. Targeted proteomics, employing techniques like selected reaction monitoring (SRM) or data extraction from data-independent acquisition digital maps, offers a promising alternative to traditional immunochemical methods, particularly in studying post-translational modifications in cancer progression. In prostate cancer, the focus is on developing biomarkers to differentiate between less aggressive and more aggressive forms of the disease, with mathematical models proposed for better predicting treatment outcomes, dosages, and schedules [32,89,90]. OMICS technologies, particularly pharmacometabolomics, contribute to predicting drug responses and optimizing dosages, as seen in organ transplantation through the monitoring of immunosuppressant drugs and their metabolites using mass spectrometry [21,91]. By correlating patients’ baseline metabolic profiles with their responses, these technologies facilitate stratification for disease susceptibility prediction and treatment with the drugs offering the most favorable therapeutic outcomes [21]. The term “Pharmacometabolomics” involves understanding drug or xenobiotic effects and predicting individual variations in drug response by analyzing both baseline metabolic profiles before treatment and the effects of drug treatment over time. This approach is powerful for predicting therapeutic responses, as metabolic changes precede phenotypic changes [92]. In precision medicine, analyzing the perturbations in low-molecular-weight endogenous and exogenous metabolite levels is crucial for selecting biomarkers to predict responses and monitor a patient’s health status during treatment [93]. Prostate cancer has been extensively studied using pharmacometabolomic approaches, focusing on understanding the metabolic changes associated with therapeutic response and castration resistance [94]. In vitro studies using PCa cell lines have examined the effects of inhibitors on metabolomic profiles, revealing alterations in amino acids, fatty acids, and other metabolites. Qu F. et al. investigated the antitumor effects of androgen receptor antagonists on PCa cell lines, highlighting significant intracellular changes [95]. Additionally, metabolic dysregulations in castration-resistant PCa were explored using in vitro and animal models, identifying the increased metabolites associated with higher energy and biosynthetic demands. Human studies, mainly in serum, plasma, and tissue, have analyzed the metabolic changes occurring under hormone therapy and chemotherapy regimens, identifying potential biomarkers for therapeutic responses and revealing the metabolic alterations associated with treatment outcomes, such as changes in bile acid metabolism, steroid synthesis, and ketogenesis. Understanding these metabolic differences can inform the development of personalized treatment strategies for PCa [94]. On the other hand, Pharmacoproteomics is a field that explores the interactions between drugs and the proteome; it involves studying how drugs influence protein expression, post-translational modifications, and interactions within a biological system, aiming to understand the molecular mechanisms of drug action, identify potential drug targets, and optimize therapeutic strategies by analyzing the proteomic changes induced by drugs. This approach provides valuable insights into drug efficacy, safety, and individualized treatment responses, contributing to developing more effective and personalized medical interventions [96]. Recent research on prostate cancer has utilized pharmacoproteomic approaches, establishing primary cell models to investigate gene mutations, mRNA/protein/surface protein distributions, and responses to pharmaceuticals. Integrated multi-omics analyses have identified potential prognostic biomarkers and therapeutic targets, contributing to more precise diagnoses and therapies [97]. The combination of Pharmacoproteomics and advanced machine learning techniques, like deep learning, represents a cutting-edge approach with significant promise for revolutionizing drug development, personalized medicine, and our overall understanding of the intricate interplay between drugs and the proteome [96]. Finally, integrating OMICS technologies into cancer research and clinical practice enhances our understanding of the disease and brings tangible benefits to personalized medicine, precise drug dosing, and improved clinical services, ultimately reducing the global healthcare burden associated with cancer. To achieve these objectives, both proteomics and metabolomics must shift their focus from discovering biomarkers to implementing a thorough validation process and applying their findings in clinical trials [32].

## 8. Clinical and Preclinical Applications of Metabolomics and Proteomics in Prostate Cancer Research

In contrast to other “omics” categories, metabolomics and proteomics hold a distinct advantage due to their proximity to the specific disease, making them more advantageous [9]. Additionally, their capacity to manipulate proteins and metabolites positions them as viable therapeutic targets, a vital factor in addressing challenges related to therapeutics for PCa. The exploration of the landscape of clinical trials utilizing metabolomics and proteomics technologies in the context of prostate cancer has gained more attention recently. Table 4 provides a comprehensive overview of these trials, offering insights into their methodologies and objectives.

### 8.1. Metabolomics to Elucidate Molecular Mechanisms

The PCa metabolome exhibits a buildup of metabolic intermediates and an upregulation of the genes involved in the tricarboxylic acid cycle. These findings align with the reactivation of mitochondrial aconitase, the restoration of metabolic flux in the Krebs cycle, and the stimulation of de novo lipogenesis and cholesterogenesis [98]. A study was conducted to investigate the molecular mechanisms underlying prostate cancer (PCa) pathogenesis through a comprehensive analysis of gene–metabolite regulatory networks and metabolic dysregulation. Utilizing gas chromatography–mass spectrometry (GC-MS) metabolomics and RNA-seq analyses in prostate tumors and matched adjacent normal tissues, the research reveals a significant accumulation of metabolic intermediates and an enrichment of genes in the tricarboxylic acid (TCA) cycle, indicating hyperactivation in PCa tissues. The study further highlights the correlation of fumarate and malate levels with Gleason score, tumor stage, and gene expression, particularly in branched-chain amino acid degradation. The findings offer a comprehensive understanding of PCa pathophysiology and suggest avenues for developing new therapeutic strategies [33]. Moreover, Ren et al. employed an integrative approach, combining transcriptomics and metabolomics analyses of 25 paired human prostate cancer tissues and corresponding noncancerous tissues. The study identified dysregulation in various pathways at both metabolic and transcriptional levels, such as cysteine and methionine metabolism, nicotinamide adenine dinucleotide metabolism, and hexosamine biosynthesis. Notably, the metabolite sphingosine exhibited high specificity and sensitivity in distinguishing prostate cancer from benign prostatic hyperplasia, particularly in patients with low prostate-specific antigen levels (0–10 ng/mL). The investigation also revealed compromised sphingosine-1-phosphate receptor two signaling, downstream of sphingosine, indicating the loss of a tumor suppressor gene and a potential oncogenic pathway suitable for therapeutic targeting. The integration of metabolomics and transcriptomics provided a comprehensive overview of the molecular perturbations in prostate cancer and initiated a preliminary exploration of a unique metabolic signature with potential applications in discriminating prostate cancer from normal tissue and benign prostatic hyperplasia. This has the potential to aid in identifying new targets for therapeutic interventions and discovering biomarkers for this condition [99].

### 8.2. Metabolomics to Uncover Resistance Mechanisms

Most prostate cancer cases progress to castration resistance, marked by rising PSA levels and metastasis. About 10–20% of prostate cancers transition to castration-resistant prostate cancer (CRPC) within five years, with 84% already having metastases at diagnosis [100]. The median survival for castration-resistant prostate cancer (CRPC) patients ranges from 15 to 36 months post diagnosis [101]. Therapy resistance in advanced prostate cancer involves androgen receptor mutations, a loss of tumor suppressor genes (e.g., p53, pTEN), and disruptions in growth factor signaling (TGF-β, IGF, VEGF). Targeting these pathways may overcome therapeutic resistance in advanced prostate cancer [102]. Genetic alterations in prostate cancer lead to distinct metabolomic changes, offering diagnostic and prognostic insights and potential therapeutic targets. Prostate tumors driven by Phosphoinositide 3-kinase (PI3K) exhibit an increase in glycolysis, aligning with the “Warburg effect” [102,103]. The metabolomic characteristics of prostate cancer include heightened Fatty Acid Synthase (FAS)-mediated fatty acid metabolism and uptake, particularly in Transmembrane Protease, Serine 2 (TMPRSS2)-Ets variant 1 (ERG) translocation-positive samples. Notably, a unique shift involves a decreased citrate concentration and increased citrate metabolite secretion, mediated by activated mitochondrial aconitase (m-aconitase), a process influenced by declining zinc levels [102]. The TMPRSS2-ERG mutation plays a vital role in the transition from pre-malignant states to prostate cancer. Fatty acid oxidation and citrate metabolism elevation contribute to increased Adenosine Triphosphate (ATP) availability in prostate cancer cells [104]. Androgen deprivation therapy (ADT) impacts metabolites like lactate and total choline, monitored through Magnetic Resonance Imaging. Metabolomic markers, including Choline Phosphate and cysteine, offer predictions of disease recurrence [34]. MicroRNA profiles, such as miR-96 and miR-21, indicate castration resistance and correlate with tumor grade. Overall, understanding and interpreting metabolomic data will unveil critical metabolites as novel biomarkers of prostate cancer progression and the emergence of therapeutic resistance. The tumor microenvironment promotes therapeutic resistance by modifying stromal components to enhance invasion, angiogenesis, and metastases. In metastatic disease, there is a shift from fibroblasts to carcinoma-associated stromal cells; thus, targeting stromal expression patterns offers a novel therapeutic approach by inhibiting the factors favoring cell differentiation [102].

### 8.3. Metabolic Phenotyping in Diagnostics

Metabolomics reveals the biochemical activities of a biological system with high sensitivity and spatial precision, thus, metabolomic profiles from biospecimens such as tissue samples, blood, urine, or cerebrospinal fluid could be utilized to elucidate the metabolic phenotype associated with a disease [105,106]. PCa cells are recognized for restructuring their cellular metabolism to fulfill increased survival, proliferation, and invasion requirements. Exploring their intricate metabolic reprogramming, a newly recognized feature of this cancer presents potential avenues for advancing cancer diagnosis, prognosis, and treatment. In this context, the integration of multi-omics data, including metabolomics, holds remarkable potential for unveiling the molecular alterations that drive the metabolic rewiring in complex diseases like prostate cancer. In recent years, metabolic phenotyping has emerged as a potent method for discovering new molecular biomarkers and identifying metabolic vulnerabilities in cancer, offering potential therapeutic opportunities. Numerous metabolomics analyses have been conducted on PCa samples to delineate the unique metabolic profile linked to PCa progression and identify potential clinical biomarkers that aid in diagnosis. These studies have revealed distinct metabolic changes distinguishing healthy and PCa samples. Comparing benign and PCa tissue samples, Lima et al. found significant dysregulations in 26 metabolites and 21 phospholipid species using NMR and MS analyses. Amino acid metabolism, glycerophospholipid metabolism, and other pathways were identified as upregulated in PCa tissues, aligning with previous studies [107,108,109]. These findings enhance our understanding of the metabolic changes associated with PCa’s development [110]. In healthy prostate cells, high zinc concentrations inhibit mitochondrial aconitase (ACO2), leading to decreased citrate oxidation and a disruption of the tricarboxylic acid (TCA) cycle. Conversely, reduced zinc levels in PCa tumors activate ACO2, restoring the TCA cycle and increasing metabolism. Metabolic studies have reported decreased citrate levels and elevated concentrations of TCA cycle intermediates in PCa tumors compared to healthy tissues, indicating increased TCA cycle metabolism. Other metabolic alterations include lower levels of polyamines, sarcosine metabolism, dysregulations of amino acids, and changes in the metabolites involved in cellular membrane metabolism.

### 8.4. Proteomics to Elucidate the Molecular Mechanisms

Chen et al. conducted a bioinformatics analysis of proteomics data to investigate prostate cancer (PCa)’s occurrence and metastasis [111]. Differentially expressed proteins (DEPs) were categorized into two groups: PCa versus benign tissues (P&B) and high versus low PCa metastatic tendencies (H&L). In the P&B group, 320 DEPs were identified, with DES being the most frequently reported. The H&L group revealed 353 DEPs, including MDH2 and MYH9, without known associations with PCa metastasis. DES was validated as differentially expressed between cancer and benign tissues. A pathway analysis highlighted protein transport, actin cytoskeleton regulation, and ECM–receptor interaction in the H&L group, presenting novel areas for investigation. The identified DEPs may be potential biomarkers for PCa detection and aggressiveness predictions. Additionally, the revealed biological processes and pathways offer insights into the molecular mechanisms of PCa’s carcinogenesis and metastasis, suggesting new avenues for clinical treatment targets [111]. Kim et al. studied the rapid proteomic changes induced by androgen treatments in VCaP cells, identifying and quantifying 5529 proteins over different time points (5, 15, 30, and 60 min). They identified five protein clusters involved in androgen-initiated signal transmission and established an androgen receptor (AR)-interacting protein network. The research provides valuable insights into the molecular mechanisms underlying castration-resistant prostate cancer (CRPC)’s progression, validated through a mouse xenograft model and patient samples [112]. 

### 8.5. Proteomics to Unravel Resistance Mechanisms

Cancer cells, including those in advanced-stage prostate cancer, show enhanced glycolytic activity, creating an acidic environment harmful to normal cells. Additionally, primary and advanced PCa exhibit increased de novo fatty acid and protein synthesis. To gain deeper insights into the molecular mechanisms driving the development of androgen resistance in prostate cancer, particularly in lethal phenotypes, a proteomics approach was employed in a study by Höti N. et al. The results underscored the heightened involvement of metabolic pathways in androgen resistance. Additionally, the study revealed an amplification of the PI3K/AKT pathway, proteasome protein overexpression, and impaired mitochondrial oxidative phosphorylation in castration-resistant LNCaP-95 cells compared to LNCaP cells. Intriguingly, Dicer, a microRNA regulator, was induced in androgen-ablated LNCaP-95 prostate cancer cells. If confirmed in clinical studies, these findings could significantly enhance our understanding of the intricate processes involved in the biochemical recurrence in prostate cancer [113]. Chang L. et al. performed a study to uncover the biomarkers and signaling pathways crucial for addressing radioresistance in prostate cancer. Utilizing a label-free LC-MS/MS proteomics approach, the research identified 309 signaling pathway proteins significantly altered between parental PCa cell lines and radioresistant (RR) PCa sublines. Nineteen proteins common among three paired PCa cell lines, associated with metastasis, progression, and radioresistance, were pinpointed. The PI3K/Akt, VEGF, and glucose metabolism pathways emerged as central in PCa’s radioresistance. These potential protein markers were validated in PCa-RR cell lines and animal xenografts, and Aldolase A (ALDOA) was selected for a radiosensitivity study. Depleting ALDOA and radiotherapy effectively reduced colony formation, induced apoptosis, and increased radiosensitivity in PCa-RR cells. These findings suggest that PCa radioresistance involves multifactorial traits, and targeting identified proteins or signaling pathways, primarily through ALDOA downregulation, in conjunction with radiotherapy holds promise for overcoming PCa’s radioresistance [114].

### 8.6. Proteomics to Identify Biomarkers

Proteomics has significantly impacted prostate cancer biomarker discovery. An ideal cancer protein biomarker should be measurable in body fluids or tissues, reflecting the cancer’s presence, stage, aggressiveness, treatment response, and recurrence likelihood. Relying on a single protein like PSA may not meet biomarker criteria; instead, a combination of multiple protein biomarkers is likely more helpful for improved PCa diagnosis and monitoring. Rifai et al. outline a four-stage process for identifying new protein biomarkers, which requires technologies for rapid and consistent identification across the disease proteome’s dynamic range. While various biological sample types can be used for proteomics-based biomarker discovery, each type (Table 5) has specific advantages and disadvantages in the search for clinically helpful protein biomarkers [5,115]. In diagnostic biomarker discovery studies, a focused strategy is occasionally used when examining fluid samples. This involves concentrating on proteins with known elevated levels in prostate tissues. TGM4, a protein highly concentrated in prostate tissues compared to non-prostate tissues, is an example. While it is not expected to be actively secreted into the blood, it can be identified in urine [115]. Another study performed by Jedinak A. aimed to identify and validate non-invasive biomarkers for distinguishing between benign prostate hyperplasia (BPH) and localized prostate cancer (PCa). Using a quantitative isobaric tags for relative and absolute quantitation (iTRAQ) LC/LC/MS/MS analysis, three proteins—β2M, PGA3, and MUC3—were identified and validated in 173 urine samples from BPH (N = 83) and PCa (N = 90) patients. A univariate analysis showed significant elevations in the urinary levels of β2M, PGA3, and MUC3 in PCa patients. A multivariate logistic regression analysis revealed AUC values ranging from 0.618 to 0.668 for individual proteins. Combining all three proteins improved diagnostic accuracy, yielding an AUC of 0.710. Further enhancement was observed when combined with PSA categories, reaching an AUC of 0.812. These findings suggest that urinary β2M, PGA3, and MUC3, individually or in combination with PSA categories, have clinical utility for noninvasively distinguishing between BPH and localized PCa [116].

## 9. Integrated Metabolomics and Proteomics

Integrating metabolomics and proteomics is essential for understanding biological systems comprehensively. The proteome’s characteristics closely resemble the metabolome’s [117]. Consequently, the combined results obtained from both domains can uncover crucial networks and signaling pathways that significantly influence the metabolic regulation of specific biological proteins. This integrated approach, in turn, plays a vital role in identifying potential targets for therapeutic interventions [9]. While Kim et al. identified the proteins encoded by 17,294 genes [118], Schroeder’s estimate suggests a range of 80,000 to 400,000 proteins, considering that one gene can encode multiple proteins [119]. In prostate cancer, proteomics is employed to investigate proteasomal degradation and abnormal metabolic processes. Many PCa studies have examined protein profiles and expression variations in localized or metastatic PCa. The process of analyzing a proteome sample involves separation techniques such as gel electrophoresis for gel-based methods and liquid chromatography (LC) or LC coupled with mass spectrometry (LC-MS) for liquid-based methods [120]. It is worth noting that the cost of implementing proteomics has restricted the number of integrated proteomics–metabolomics studies in the literature, especially when compared to genomics–metabolomics or transcriptomics–metabolomics studies [121]. However, recent developments in proteome mapping and the emergence of top–down proteomics have made this approach more feasible [117]. Integrating proteomic and metabolomic data primarily focuses on profiling, pathway mapping, and association studies. For instance, distinguishing between PCa and normal prostate cells is achieved through a combination of proteomics and metabolomics, which enables the analysis of dysregulated lipid metabolism and increased protein phosphorylation [122]. The evolution of computing capabilities has allowed this integrated approach to move beyond simple pathway mapping [121]. In the future, through the integration of various metabolomic datasets and advancements in AI methodologies, novel biomarkers for early cancer detection could be discovered [123]. Table 6 provides a comprehensive overview of significant recent studies (conducted between 2023 and 2024) employing metabolomics and proteomics to investigate various aspects of prostate cancer (PCa).

## 10. Limitations and Challenges

In contemporary clinical research, omics studies provide a dynamic set of tools for collecting and analyzing biological samples, offering innovative approaches. Nevertheless, despite the various omics applications discussed in this review, it is crucial to acknowledge that these techniques have inherent limitations. Differentiating specific cell signals o those associated with cancer and employing accurate technologies for profiling heterogeneous tumors remain significant challenges. While there have been notable advancements, mastering metabolomics and proteomics remains a complex endeavor due to the diverse nature of metabolites and their intricate characteristics, which add complexity to the metabolome. Obtaining a comprehensive overview necessitates the utilization of multiple analytical platforms for the extraction, detection, and quantification of metabolites and proteins. It is imperative to establish standardized practices to mitigate variability across different laboratories. The dynamic nature of metabolomes and proteomes, along with their susceptibility to external influences, underscores the importance of employing diverse analytical strategies and utilizing various mass spectrometry analyzers to attain a comprehensive understanding of them [1,12,24,42,82]. 

## 11. Conclusions and Future Perspectives

Metabolomics and proteomics, emerging as prominent omics sciences in biomedical research, have become invaluable tools for biomarker discovery, disease diagnosis, prognosis, and therapeutic innovation, particularly in cancer. Choosing the appropriate analytical technique (MS or NMR) depends on the analyte’s characteristics and the study’s specific goals, considering each method’s distinct features, benefits, and limitations. These techniques generate vast volumes of high-quality data requiring comprehensive analyses and processing. This begins with data pre-processing, involving techniques like normalization, alignment, and noise reduction to create readily interpretable datasets. Statistical analysis, encompassing univariate, multivariate, or a combination of both analysis, is instrumental in extracting physiologically meaningful insights, contributing to a holistic understanding of biological processes. The field of omics is in a state of constant evolution, with exciting prospects such as the development of biosensors based on specific biomarkers for early disease detection. Moreover, ongoing advancements in instrumentation propel the field forward by providing sophisticated data that allow us to gain a deeper understanding of systems biology and refine targeted medication therapies. Omics’ future lies in translating research findings into clinical applications, promising a revolution in healthcare with tailored diagnostic and therapeutic solutions.

## Figures and Tables

**Figure 1 ijms-25-05071-f001:**
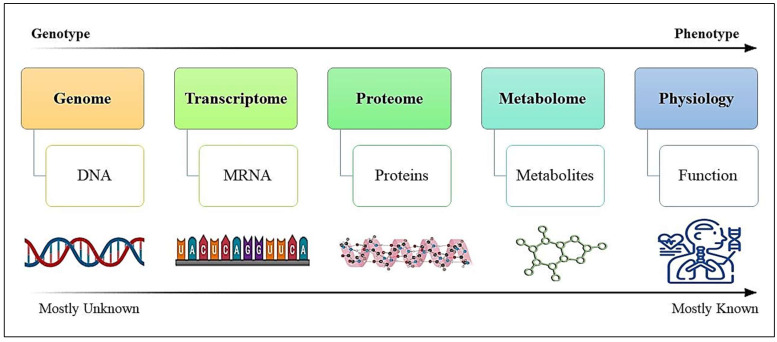
The stages of hierarchical cellular organization involved in the transition from genotype to phenotype.

**Figure 2 ijms-25-05071-f002:**
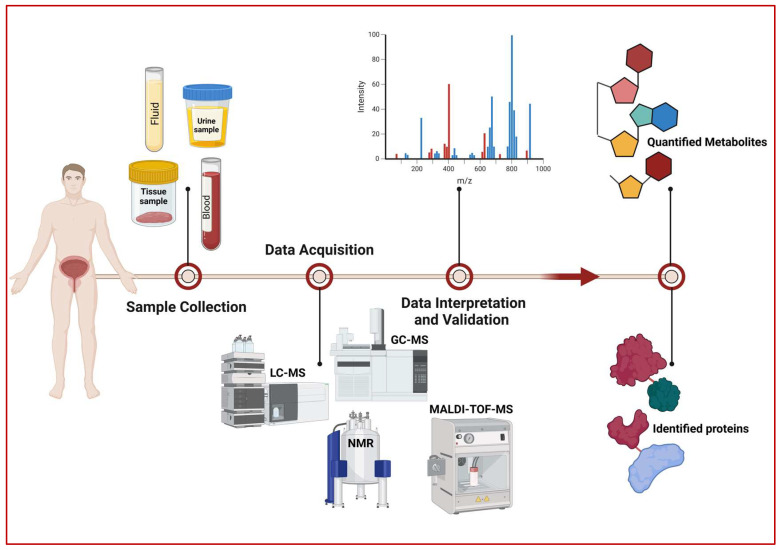
A general schematic representation of the metabolomics and proteomics’ workflow. Created with BioRender.com.

**Figure 3 ijms-25-05071-f003:**
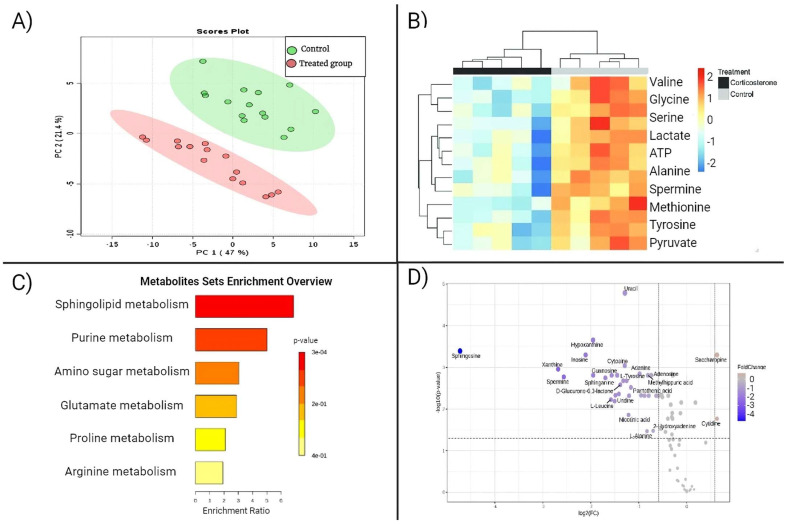
Some graphical visualization features of MetaboAnalyst 5.0. (**A**) PCA analysis plot. (**B**) Heatmap showing the differential metabolites in the statistical analysis function of MetaboAnalyst 5.0. (**C**) Pathway enrichment overview in MetaboAnalyst 5.0. Color shade is based on the *p*-value. (**D**) Volcano plot of a differential analysis in MetaboAnalyst 5.0. Created with BioRender.com.

**Figure 4 ijms-25-05071-f004:**
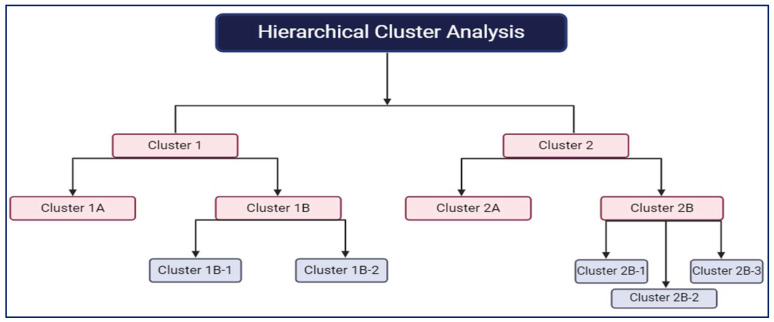
Diagrammatic representation of hierarchical clustering. Created with BioRender.com.

**Table 1 ijms-25-05071-t001:** A comprehensive overview of various biomarkers.

Biomarker	Type	Applications/Characteristics	References
PSA (Prostate-Specific Antigen)	Diagnostic/Disease Monitoring	- Widely used in PCa management, but with limited sensitivity. Elevated blood PSA levels (>4.0 ng/mL) are often indicative of prostate cancer, though the threshold’s accuracy is debated. - An androgen-dependent serine protease with various applications in PCa management.	[36,37]
hK2 (Human Glandular Kallikrein 2)	Diagnostic	- A serine protease like PSA aids in converting pro-PSA to active PSA. - Often considered alongside PSA, but cross-reactivity issues exist.	[36,38]
ANAX 3 (Annexin 3)	Diagnostic and Prognostic	- A calcium- and phospholipid-binding protein with complex roles in cell processes and tumorigenesis. - Its consistency as a PCa biomarker is debated.	[36,39]
β2M (Beta-2-Microglobulin)	Diagnostic	- A low-molecular-weight protein present on the surface of all nucleated cells in the body, it is part of histocompatibility class 1 (MHC class 1). - Associated with various cancers, including PCa. Altered MHC expression is linked to immune evasion in cancer, and β2M might help distinguish benign prostate hyperplasia (BPH) from PCa.	[36]
MSMB (Microseminoprotein-Beta)	Diagnostic	- Produced in the prostate gland with PSA, its levels decrease with PCa progression, possibly acting as a tumor suppressor. - Lower serum and urinary MSMB were observed in PCa compared to healthy controls.	[36,40]
SAA (Serum Amyloid A)	Prognostic and Predictive	- A liver-produced lipoprotein with implications in inflammation and cancer, explored for its prognostic and therapeutic value in various cancers, including PCa.	[36,41]
EN2 (Engrailed-2)	Diagnostic	- Part of the HOX gene family and studied as a urinary biomarker. - Potential indicator of PCa volume.	[36]
Myo-inositol, Citrate, Polyamine Spermine	Prognostic	- A metabolic biomarker indicative of prostate cancer.	[42]
Sarcosine	Predictive	- Metabolic biomarker connected to prostate cancer. - Non-invasive biomarker potential due to substantial increase in urine, tissue, and plasma during advanced PCa stages, including metastasis. - Intracellular sarcosine levels elevated in invasive prostate cancer cell lines compared to benign prostate epithelial cells, indicating its role in cancer progression. - Elevated in advanced stages, it surpasses traditional PSA markers, aiding in low-grade cancer detection and therapy selection. Its regulatory role in cancer progression and validation as an oncometabolite emphasizes its importance in prostate cancer management.	[43,44,45,46,47,48,49,50,51,52]
Kynurenine	Predictive	- Detectable in plasma, urine, and tissue, linked to prostate cancer development and recently associated with cancer aggressiveness based on Gleason scores.	[45,53]
Choline	Predictive	- Implicated in cancer development and DNA repair modulation, functions as a predictive biomarker for prostate cancer when levels exceed 10 mol/L in contrast to healthy individuals.	[55,56]
Exosomes	Diagnostic and Predictive	- Exosomal mRNA plays a crucial role in diagnosing, treating, and predicting prostate cancer outcomes. It influences tumor microenvironment dynamics and cancer aggressiveness. - Unique prostate-cancer-specific content is present in blood and urine EVs. - Exosome-isolated proteins and urinary exosomes show promise as non-invasive biomarkers.	[57,58,59,60,61,62,63,64,65]

**Table 3 ijms-25-05071-t003:** A comprehensive comparison of key characteristics between (MS) and (NMR) techniques.

Characteristic	MS	NMR	References
Sensitivity	High	Low	[24,25]
Reproducibility	Moderate	Very high	
Type of sample	Liquid	Liquid and tissue	
Sample preparation	Complex	Simple	
Sample recovery	Destructive	Non-destructive	
Analyte identification	Easy	Difficult	
Metabolite database	Only uniform for GC-MS	Uniform	
Number of known identifiable metabolites	Thousands of metabolites in a single measurement	Hundreds of metabolites in a single measurement	
Common techniques	GC-MS, LC-MS	H&C NMR, 2D-NMR	
Quantification	Not fully quantitative without appropriate standards	Quantitative	

**Table 4 ijms-25-05071-t004:** Some clinical trials employing metabolomics and proteomics technology in prostate cancer (adopted from clincaltrials.gov, accessed on 22 November 2023).

Study Title	Aim and Intervention	Country	ID	Status
A Multi-omics Study of Metastatic Prostate Cancer (MOSMPCA)	Multi-omics technologies, including proteomic methods, investigating the mechanism underlying tumor progression, identifying distinct subtypes of tumors, and discovering potential treatment targets in patients diagnosed with metastatic prostate cancer.	China	NCT04660617	Unknown ^†^
Decipher Lethal Prostate Cancer Biology—Urine Metabolomics	Developing urine metabolomics markers to improve prostate cancer screening and prevention while reducing the overtreatment of insignificant cases.	Taiwan	NCT03237702	Recruiting
Predictive and Prognostic Markers for Treatment Outcomes in Prostate Cancer Patients	Leveraging urine metabolomics and proteomics profiling to identify predictive and prognostic markers for treatment outcomes in prostate cancer patients.	Taiwan	NCT03237026	Recruiting
UCI 03-72 Prostate Cancer Detection by Serum Proteomic Profiling	Investigators use serum proteomic profiling to assess the predictive capabilities of SELDI (Surface Enhanced Laser Desorption Ionization) analysis compared to PSA tests in determining biopsy outcomes.	California, United States	NCT00355758	Terminated (lack of personnel)
Study in Predicting Outcome of Patients Undergoing Radiation Therapy for Prostate Cancer	A diagnostic trial is underway to investigate blood and urine proteins as potential predictors of treatment outcomes in patients undergoing radiation therapy for prostate cancer.	Maryland, United States	NCT00045331	Completed
Study of Blood and Urine Samples in Patients With Newly Diagnosed Localized Prostate Cancer Treated With Hormone Therapy and Radiation Therapy	In this clinical trial, blood and urine samples from patients with newly diagnosed localized prostate cancer are being studied to explore the impact of hormone therapy and radiation therapy on treatment outcomes using proteomics profiling.	Dublin, Ireland	NCT00955435	Unknown ^†^
Metabolic Impact of Prospective Controlled Mediterranean-Type Diets on Prostate Cancer	To investigate the influence of Mediterranean-type diets on the metabolism of men diagnosed with localized prostate cancer.	Ohio, United States	NCT05590624	Not yet recruiting

^†^ Study has passed its completion date, and its status has not been verified in more than two years.

**Table 5 ijms-25-05071-t005:** Sample selection for biomarker discovery. Table adapted from [5].

	Tissue Biopsy, Needle biopsy	Serum and Plasma	Urine	Prostatic Fluid and Seminal Plasma
Advantages	Direct analysis of tumor protein expression/activation	Non-invasive collection	Non-invasive collection	Slightly invasive collection
	Diagnostic markers	Fast and low-cost sample preparation	High volume	Abundant in prostate-derived proteins
	Prognostic markers	Diagnostic markers	Rich in prostate-derived proteins	Quick and low-cost sample preparation
	Most useful for patient stratification in terms of response to therapy	Prognostic markers	Fast and low-cost sample preparation	Diagnostic markers
			Diagnostic markers	Prognostic markers
			Prognostic markers	
Limitations	Invasive collection	Low abundance of potential biomarkers	Low abundance of potential biomarkers	Minimal abundance of potential biomarkers
	Limited quantity	Dynamic concentration range	Dynamic concentration range	Dynamic concentration range
	Must be snap-frozen within 30 min of collection	Intra- and inter-patient variability in composition	Intra- and inter-patient variability in composition	Intra- and inter-patient variability in composition
	Complex sample preparation Tissue sampling errors		Variability in sample collection	

**Table 6 ijms-25-05071-t006:** A comprehensive overview of significant recent studies (conducted between 2023 and 2024) investigating various aspects of prostate cancer (PCa).

Title	Aim	Methods	Results	Reference
Alterations of plasma exosomal proteins and metabolites in castration-resistant prostate cancer (CRPC) progression	Characterizing the proteomic and metabolomic profiles of exosomes to assess their diagnostic potential in prostate cancer (PCa), particularly CRPC. Investigating the functional roles of specific exosome biomarkers in CRPC progression.	Conducted integrated proteomics and metabolomics analysis of plasma-derived exosomes from tumor-free controls (TFC), PCa, and CRPC patients.	1. Leucine-rich alpha2-glycoprotein 1 (LRG1) and inter-alpha-trypsin inhibitor heavy chain H3 (ITIH3) identified as potential predictive markers for CRPC. 2. Differential metabolite expression in exosomes distinguished PCa from TFC and CRPC. 3. LRG1 protein significantly upregulated in advanced prostate cancer. 4. Exosomes overexpressing LRG1 derived from PCa cells notably enhanced angiogenesis.	[124]
Prediction of clinically significant prostate cancer through urine metabolomic signatures: A large-scale validated study	Developing urine tests to predict clinically significant PC (sPC) in men at risk.	Analyzed urine samples from 928 men, including 660 PC patients and 268 benign subjects, using GC/Q-TOF MS metabolomic profiling to construct four predictive models.	1. Models I, II, III, and GS, involving 26, 24, 26, and 22 metabolites, respectively, augmented by five clinical risk factors, significantly improved AUCs, aiding in sPC prediction. 2. The combined urine test of metabolic markers and clinical factors effectively predicted sPC, guiding biopsy necessity for men with an elevated PC risk.	[125]
Radiotherapy induces innate immune responses in patients treated for prostate cancers	Investigating the innate immune responses induced by radiotherapy in patients with prostate cancer.	Investigated systemic clinical responses post prostate stereotactic body radiotherapy using proteomic and metabolomic analyses.	1. Observed increased DNA damage response and persistent innate immune signaling in patients after prostate stereotactic body radiotherapy. 2. Observed differential immune responses and metabolite profiles between patients in remission and those experiencing disease progression post radiotherapy.	[126]
Metabolomic profiles of intact tissues reflect clinically relevant prostate cancer subtypes	For improved treatment stratification, reliable approaches are needed to faithfully differentiate between high- and low-risk tumors and to predict therapy response at diagnosis.	Applied a metabolomic approach based on HR-MAS NMR to analyze intact biopsy samples obtained from patients treated by prostatectomy, combined with advanced statistical methods to identify metabolomic profiles reflecting tumor subtypes based on Ki67 and PSA immunoreactivity.	Identified distinct metabolite patterns reflecting clinically relevant prostate cancer subtypes based on Ki67 and PSA immunoreactivity, including alterations in choline, phosphocholine/glycerophosphocholine, glycine, creatine, glutamate/glutamine, taurine, and lactate.	[127]
Integrating intracellular and extracellular proteomic profiling for in-depth investigations of cellular communication in a model of prostate cancer	Exploring cellular communication in prostate cancer through integrated intracellular and extracellular proteomic profiling.	Employed cellular-based proteomics to comprehensively profile both intracellular and extracellular proteomes in a prostate cancer model, enabling investigations into cellular communication dynamics.	1. Revealed over 8000 proteins through intracellular and extracellular proteomic profiling, shedding light on cellular communication dynamics in prostate cancer’s development and progression. 2. Demonstrated the utility of integrated intracellular and extracellular proteomic profiling for investigating the cellular communication dynamics in prostate cancer.	[128]
Serum organic acid metabolites as potential biomarkers for prostatitis, benign prostatic hyperplasia, and prostate cancer	Identifying serum organic acid metabolites as potential biomarkers for distinguishing prostatitis, benign prostatic hyperplasia (BPH), and prostate cancer (PCa).	Employed untargeted and targeted LC-MS to identify and verify serum organic acid metabolites in patients with prostatitis, BPH, and PCa, enabling the development of diagnostic models for disease differentiation.	1. Identified specific serum organic acid metabolites with good sensitivity and specificity for differentiating prostatitis, BPH, and PCa, including phenylacetic acid, pyroglutamic acid, citric acid, malic acid, D-glucuronic acid, and others. 2. Highlighted the potential of serum organic acid metabolites as biomarkers for differentiating between prostatitis, BPH, and PCa, offering diagnostic insights into these conditions.	[129]
Integrative analysis of transcriptomic and metabolomic profiles reveals enhanced arginine metabolism in androgen-independent prostate cancer cells	Investigating enhanced arginine metabolism in androgen-independent prostate cancer cells through integrative analysis of transcriptomic and metabolomic profiles.	Through RNA sequencing and LC-MS/MS analysis, integrating transcriptomic and metabolomic data for a comprehensive understanding.	1. Identified enhanced arginine metabolism in androgen-independent prostate cancer cells, with the arginine and proline metabolism pathway commonly altered at both transcriptional and metabolic levels, suggesting its substantial association with CRPC. 2. Emphasized the substantial association between the arginine and the proline metabolism pathway and CRPC, underlining the importance of targeting dysregulated metabolites and differentially expressed genes for clinical management.	[130]
Relationship between 4-Hydroxynonenal (4-HNE) as systemic biomarker of lipid peroxidation and metabolomic profiling of patients with prostate cancer	Investigating the association between 4-Hydroxynonenal (4-HNE) as a systemic biomarker of lipid peroxidation and metabolomic profiling in patients with prostate cancer.	Utilizing immunohistochemistry, plasma sample analysis, and LC-ESI-QTOF-MS and GC-EI-Q-MS metabolomic techniques.	1. Revealed the absence of 4-HNE-protein adducts in prostate carcinoma tissue but increased 4-HNE-protein levels in the plasma of these patients, along with altered metabolomic profiles indicating a positive association of different long-chain and medium-chain fatty acids with the presence of prostate cancer and an affected unsaturated fatty acids biosynthesis pathway. 2. Revealed that altered lipid metabolism and the unsaturated fatty acids biosynthesis pathway are associated with increased 4-HNE plasma protein adducts in prostate cancer patients.	[131]

## Data Availability

Not applicable.

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
