# Peer review of "Metabolomics and Proteomics in Prostate Cancer Research: Overview, Analytical Techniques, Data Analysis, and Recent Clinical Applications"

_ijms, 2024, doi:10.3390/ijms25105071_

Round 1

Reviewer 1 Report

Comments and Suggestions for Authors

Dear Authors,

the manuscript is a review of the current methods of omics in PCa. It reports many different methods of studying omics and its current applications.

Major issues

The article it's mainly descriptive, but current results available should emerge more. It would be nice to add a table reassuming results of concluded trials if available

Minor issue

Table 4 is displayed after Table 5. It would be suitable to show the tables just after their paragraph

Author Response

Dear Editor, IJMS

Thank you for your efforts thus far,

Please find Attached the revised version of our manuscript.

We truly appreciate the constructive criticism and encouraging comments by the reviewers. The manuscript has been considerably revised in accordance with the comments and we believe this has significantly improved the quality of our manuscript. We used track changes for the revision.

To reviewer 1: Thank you very much for taking the time to review this manuscript. Please find our responses to specific comments and point-by-point details about the revisions below:

The manuscript is a review of the current methods of omics in PCa. It reports many different methods of studying omics and its current applications.

Point 1: Major issues: The article it's mainly descriptive, but current results available should emerge more. It would be nice to add a table reassuming results of concluded trials if available.

Response 1: Thank you for your insightful feedback. We acknowledge the importance of emphasizing the current results in our manuscript. To address this concern, we have added Table 6 to provide a comprehensive overview of significant studies conducted between 2023 and 2024. This table encompasses various aspects of prostate cancer research utilizing metabolomics and proteomics techniques. We believe this addition will greatly enhance the clarity and relevance of our discussion on current findings in the field, significantly improving the depth and relevance of our article.

Point 2: Minor issue: Table 4 is displayed after Table 5. It would be suitable to show the tables just after their paragraph.

Response 2: Thank you for your feedback. We appreciate your attention to detail. To address your suggestion, we have moved Table 4 to appear directly after its corresponding paragraph as requested. We believe this adjustment will enhance the organization and readability of our manuscript.

Reviewer 2 Report

Comments and Suggestions for Authors

The review article entitled “Metabolomics and Proteomics in Prostate Cancer Research: Overview, Analytical Techniques, Data Analysis and Recent Clinical Applications” by Daffaie et al., is very well written except for minor grammatical and language errors. Authors should correct those minor errors in the manuscript. In section 3, or in a separate section authors should include exosome as a biomarker for prostate cancer. There is a myriad literature available on exosomes as a promising diagnostic model for prostate cancer. Recent article by Joshi S. et al., have underlined significance of metabolite signatures from exosomes to predict response in breast cancer patients. A similar approach can be employed to prostate cancer. Authors should consider including this section in their review article and cite relevant articles including Joshi S, et al., Exosomal Metabolic Signatures Are Associated with Differential Response to Neoadjuvant Chemotherapy in Patients with Breast Cancer. Int J Mol Sci. 2022 .

Author Response

Dear Editor Prof. Maurizio Battino,
IJMS
Thank you for your efforts thus far,
Please find Attached the revised version of our manuscript.
We truly appreciate the constructive criticism and encouraging comments by the reviewers. The manuscript has been considerably revised in accordance with the comments and we believe this has significantly improved the quality of our manuscript. We used track changes for the revision.
To reviewer 2: Thank you very much for taking the time to review this manuscript. Please find our responses to specific comments and point-by-point details about the revisions below:
Point 1: The review article entitled “Metabolomics and Proteomics in Prostate Cancer Research: Overview, Analytical Techniques, Data Analysis and Recent Clinical Applications” by Daffaie et al., is very well written except for minor grammatical and language errors. Authors should correct those minor errors in the manuscript.
Response 1: Thank you for your feedback. We are pleased to hear that you found it well-written. We acknowledge the presence of minor grammatical and language errors and have taken steps to address them. Grammarly was employed to thoroughly resolve these issues throughout the manuscript. We trust that these corrections have enhanced the overall clarity and professionalism of our work.
Point 2: In section 3, or in a separate section authors should include exosome as a biomarker for prostate cancer. There is a myriad literature available on exosomes as a promising diagnostic model for prostate cancer. Recent article by Joshi S. et al., have underlined significance of metabolite signatures from exosomes to predict response in breast cancer patients. A similar approach can be employed to prostate cancer. Authors should consider including this section in their review article and cite relevant articles including Joshi S, et al., Exosomal Metabolic Signatures Are Associated with Differential Response to Neoadjuvant Chemotherapy in Patients with Breast Cancer. Int J Mol Sci. 2022.
Response2: Thank you for your valuable input regarding the inclusion of exosomes as a biomarker for prostate cancer in our review article. We agree that this is an important aspect that warrants attention. In response to your suggestion, we have added a new paragraph discussing the use of exosomes as biomarkers in prostate cancer within section 4 of the manuscript. Additionally, we have cited the relevant article by Joshi et al., titled "Exosomal Metabolic Signatures Are Associated with Differential Response to Neoadjuvant Chemotherapy in Patients with Breast Cancer," as you recommended. Furthermore, we have incorporated one relevant study into Table 6 for comprehensive coverage. We appreciate your insightful recommendation and believe that these additions enrich the scope and relevance of our review article.

Reviewer 3 Report

Comments and Suggestions for Authors

The study "Metabolomics and Proteomics in Prostate Cancer Research: Overview, Analytical Techniques, Data Analysis, and Recent Clinical Applications" is an impressive work that highlights the importance of omics techniques in the exploration of prostate cancer. The authors have shown great dedication and attention to detail in their manuscript. However, to make it even better, certain aspects, such as the introduction and the length of the manuscript, could benefit from some refinement before publication.

Strengthen the introduction by clearly stating the research problem(more focus on the importance of OMICS in PCa and its relevance).

Consider revising sections to eliminate redundant information and focus on essential points. 

Author Response

Dear Editor Prof. Maurizio Battino,
IJMS
Thank you for your efforts thus far,
Please find Attached the revised version of our manuscript.
We truly appreciate the constructive criticism and encouraging comments by the reviewers. The manuscript has been considerably revised in accordance with the comments and we believe this has significantly improved the quality of our manuscript. We used track changes for the revision.
To reviewer 3: Thank you very much for taking the time to review this manuscript. Please find our responses to specific comments and point-by-point details about the revisions below:
Point 1: The study "Metabolomics and Proteomics in Prostate Cancer Research: Overview, Analytical Techniques, Data Analysis, and Recent Clinical Applications" is an impressive work that highlights the importance of omics techniques in the exploration of prostate cancer. The authors have shown great dedication and attention to detail in their manuscript. However, to make it even better, certain aspects, such as the introduction and the length of the manuscript, could benefit from some refinement before publication.
Response 1: Thank you for your kind words regarding our study, We greatly appreciate your recognition of our dedication and attention to detail. In response to your suggestions, we have made several improvements. Firstly, we have shortened some sentences throughout the manuscript to enhance readability and clarity. Additionally, we have removed a subheading from the introduction to prevent it from becoming overly lengthy, thereby improving the flow of the introduction section. We believe that these refinements have enhanced the overall quality of the manuscript, and we are grateful for your constructive feedback.
Point 2: Strengthen the introduction by clearly stating the research problem(more focus on the importance of OMICS in PCa and its relevance).
Response2: Thank you for your suggestion to strengthen the introduction, we have addressed this comment as requested, ensuring that the introduction now provides a clear and concise
statement of the research problem while emphasizing the significance of OMICS techniques in the study of PCa.
Point 3: Consider revising sections to eliminate redundant information and focus on essential points.
Response 3: Thank you for your valuable feedback. We appreciate your suggestion to revise sections to eliminate redundant information and focus on essential points. After a thorough review of the manuscript, we have made the necessary revisions to streamline the content. We aimed to ensure that each section emphasizes the most critical points while eliminating redundancy. We believe these revisions have indeed improved the clarity and effectiveness of the manuscript.
